# Sickness absence and disability pension trajectories among individuals on sickness absence due to stress-related disorders. Two prospective population-based cohorts with 13-month follow-up

**Katalin Gémes** [ID]*, **Emma Pettersson, Sara Sjölund Andoff, Kristin Farrants** [ID],
**Emilie Friberg, Kristina Alexanderson**

Department of Clinical Neuroscience, Division of Insurance Medicine, Karolinska Institutet, Stockholm,
Sweden

* katalin.gemes@ki.se

GERMANY

**Data Availability Statement:** The used data cannot
be made publicly available due to privacy

## Abstract

### Background

Stress-related disorders are common diagnoses for sickness absence (SA) and disability
pension (DP) in many Western countries. Knowledge on future SA/DP trajectories among
those starting such a SA spell is limited. The aims were to identify future SA/DP days trajec-
tories among individuals starting an SA spell due to stress-related disorder and investigate
socio-demographic and morbidity characteristics associated with specific trajectories.

### Methods

Using microdata from nationwide registers, we established two cohorts of all living in Swe-
den who started a new SA spell >14 days due to stress-related disorder in 2011 (N =
32,417) or in 2018 (N = 65,511), respectively. Group-based trajectory models were used to
identify trajectories of monthly average SA/DP days during the following 13 months, sepa-
rate for each cohort. We used multinomial logistic regression to investigate the associations
between sociodemographic and morbidity-related predictors and trajectory membership.

### Results

We identified six SA/DP trajectories in the two cohorts: *steep drop* (30.6% and 35.9% of all
included in 2018 and 2011); *constant fluctuating* (8.7%, 11.2%); *fast decrease* (25.5%,
24.4%); *medium decrease* (18.1%, 13.1%); *slow decrease* (10.8%, 7.3%), and *constant
high* (6.2%, 8.0%). The distributions of sociodemographic factors, multi-morbidity, and his-
tory of SA/DP differed between the trajectory groups. For example, compared to the *steep
drop* trajectory, individuals in the other trajectories were more likely to be a woman, older,
having had prior SA/DP or specialized outpatient healthcare visits.

regulations. According to the General Data Protection Regulation, the Swedish law SFS 2018:218, the Swedish Data Protection Act, the Swedish Ethical Review Act, and the Public Access to Information and Secrecy Act, these types of sensitive data can only be made available for specific purposes that meets the criteria for access to this type of sensitive and confidential data as determined by a legal review. Contact email: imas-cns@ki.se.

**Funding:** This study was financially supported by the Research Council for Health, Working Life and Welfare and the Swedish Social Insurance Agency. We utilised data from the REWHARD consortium supported by the Swedish Research Council (grant no. 2021-00154). The funders had no role in study design, data collection and analysis, decision to publish, or preparation of the manuscript.

**Competing interests:** The authors have declared that no competing interests exist.

## Conclusions

In these two explorative, population-wide cohorts, we identified six different trajectories of SA/DP days among all with a new SA spell with stress-related disorders. The trajectory groups differed regarding both sociodemographic and health-related covariates.

## Introduction

Mental disorders may cause temporary or permanent work incapacity, leading to sickness absence (SA) and/or disability pension (DP) [1–3]. In the OECD countries, the proportion of individuals with long-term SA (defined as longer than 90 days) and DP and the number of SA days due to a mental diagnosis has increased in the last twenty years [1,2]. In Sweden, mental disorders are among the most common SA diagnoses [3], and among the mental diagnoses, stress-related disorders are one of the most frequent, contributing to a vast number of SA days [2,4]. The incidence of SA due to stress-related disorders has doubled in Sweden since 2010, and has also increased in other OECD countries [2], causing considerable societal and economic burden due to loss of productivity, increase in healthcare use, and other associated social costs [5,6].

Stress-related disorders (code of F43 according to the 10th version of the International Classification of Diseases (ICD-10)) include both acute and chronic maladaptive stress-related conditions, such as acute stress reaction, posttraumatic stress disorder, and other specified and unspecified reactions to severe stress [7]. It also includes exhaustion disorder, a reaction to prolonged psychosocial stress [8]. The duration of SA spells due to stress-related disorders varies but is often long [9,10]. In a Swedish population-wide cohort from 2010–2012, 75% of the SA spells due to stress-related disorders were shorter than 180 days, while 10% of them lasted longer than 365 days [10]. Long SA is associated with both adverse individual and societal outcomes [11–14]. On an individual level, long SA has been shown to be associated with a higher risk of prolonged or new morbidities, disability pension (DP), and premature death independent from the baseline morbidities, or familiar confounding in twin studies [12–16]. Long SA also increases the risk of labor market marginalization and social isolation, which can further worsen mental health [17].

Several risk factors were identified in association with the SA duration as well as the chance of returning to work after SA due to stress-related disorders separately or together with other common mental disorders, such as anxiety and depression in previous research, including higher age, lower education, lower socioeconomic position, worse general health status, comorbid mental and somatic disorders, as well as bullying and lower social support at work [1,18–21]. However, these studies mostly conceptualized SA and return to work as a dichotomous outcome [1,19–21], which limits the possibility of investigating the diversity of SA trajectories that individuals might follow and identifying subgroups following similar SA patterns and their risk factors in the population. Other population-based studies that used trajectory analysis conducted their analysis in different study populations, such as individuals starting sick leave due to depression, which might differently affect work capacity and SA/DP trajectories compared to stress-related [22]. Investigating the diversity of SA patterns and identifying vulnerable subgroups is important for more targeted prevention and treatment efforts.

Furthermore, many of the previous studies were survey-based with self-reported outcomes, often based on selected samples, which might limit generalizability to other populations and introduce bias if missingness during sampling and follow-up is not random. Using nationwide

high-quality register data allows one to investigate trajectories in the whole population, avoid selection bias, and capture patterns that might be missed in smaller, more selected survey data. Finally, despite the strong increase in SA due to stress-related disorders in Europe during the last two decades, there have been no studies that investigate how SA patterns and risk factors that are associated with the different SA patterns changed over this period.

Therefore, we aimed to identify future SA/DP trajectories and investigate sociodemographic, work- and morbidity-related characteristics associated with different trajectory memberships in two population-wide cohorts in Sweden, one from the beginning of the increasing trend in SA due to stress-related disorder and one from a more recent period.

## Materials and methods

Two prospective cohort studies were conducted, based on pseudonymized microdata linked at individual level from four Swedish nationwide administrative registers, namely: the Longitudinal Integration Database for Health Insurance and Labour Market Studies (LISA) held by Statistics Sweden, regarding data on socio-demographics [23], the Micro-data for analyses of social insurance (MiDAS) held by the Social Insurance Agency, with information on SA and DP benefits (dates, extent (full- or part-time), and diagnoses); the National Patient Register [24] and the Cause of Death Register [25] held by the National Board of Health and Welfare, regarding inpatient and specialized outpatient (i.e., secondary) healthcare and mortality, respectively. Record linkages were made by Statistic Sweden using the unique personal identity number (PIN) assigned to all people resident in Sweden [26].

### Study populations

We formed two population-based cohorts. The first, named Cohort-2011, included all 32,520 individuals registered as living in Sweden in December 2010, aged 18–64 years, and who in 2011 began a new SA spell >14 days due to stress-related disorders (ICD-10 code F43). The second cohort, named Cohort-2018, had the same inclusion criteria, however, for the years 2017 and 2018, respectively, and included 65,785 individuals. All individuals in the two cohorts were followed prospectively for 13 months (defined by 30-day periods) from day 15 of the index SA spell regarding the sum of net days of SA/DP per month. This included days for new SA/DP during follow-up, even if the index spell had ended during the follow-up (not including the first 14 days of the new SA spells). We excluded those who died during the follow-up (n = 38 in Cohort-2011 and n = 109 in Cohort-2018) or emigrated during 2011–2012 (n = 65) or 2018–2019 (n = 165), respectively. After these exclusions, the two cohorts included 32,417 and 65,511 individuals, respectively. The data were accessed for analysis in January 2017 for Cohort-2011 and May 2024 for Cohort-2018.

### The public sickness absence and disability pension insurance system in Sweden

All residents in Sweden with income from work, unemployment, parental leave benefits, or student finance can, from the age of 16 years, receive SA benefits if their work capacity is reduced due to disease or injury [27]. For employees, the employer pays SA benefits for the first 13 days of a SA spell after the first qualifying day. A medical certificate is required from day eight. After the 14th day, the Social Insurance Agency pays the benefits, covering approximately 80% of lost income up to a certain level. All residents aged 19–64 years with a long-term or permanently reduced work capacity due to disease or injury can be granted DP. Both SA and DP may be granted full- or part-time (25%, 50%, 75%) of ordinary working hours. This means that people can have both part-time SA and DP benefits at the same time.

Therefore, we calculated SA/DP net days; e.g., two days of part-time absence for 50% were summed to one net day. All the below information on days concerns net days. The first 14 days of SA spells were not included.

## Covariates

The following covariates were included, and categorized as follows: *Sex* (Women, Men); *Age* (18–30, 31–40, 41–50, 51–64 years); *Country of birth* (Sweden, Other Nordic countries, EU-25 without Nordic countries, Rest of the world (missing was excluded)); *Educational level* (Compulsory school (≤9 years and missing information on education, n = 57 in 2011 and n = 182 in 2018), Upper secondary school/high school (10–12 years), College/university (>12 years)); *Type of living area* (Large cities: Stockholm, Gothenburg, or Malmö, Medium-sized cities; >90,000 inhabitants within 30 km from city centre, Town: <90,000 inhabitants within 30 km from city centre); *Family situation* (Married/cohabitant without children <18 years at home, Married/cohabitant with children at home, Single without children at home, Single with children at home); *Type of occupation* (White collar: legislators, senior officials and managers, professionals and technicians, Blue collar: clerks, service workers and shop and market sales workers, skilled agricultural and fishery workers, craft and related trades workers, plant and machine operators and assemblers, elementary occupations, constructed using the one digit code of categorizing major occupational areas in the Swedish version of the International Standard Classification of Occupations; *Employment status* at the beginning of the index spell (In paid work, Unemployed, On parental leave, Student); *Extent of SA* at day 15 of the spell (Full time: 100%, Part-time: 25%, 50%, or 75%); *Prior SA days* (categorised as 0, 0.25–50, 50.25–90, 90.25–180, or 180.25- net days of SA in the 12 months before the start date of the index spell) due to stress-related disorder (ICD-10: F43), due to other mental disorders (ICD-10: F00-F99 or Z73.0, excluding F43), or due to any somatic disorder (ICD-10: A00-Z99, excluding F00-F99, O80, and Z00-Z99); *Disability pension* in the 12 months before the index spell date (Yes, No); and *Specialized outpatient and inpatient healthcare use* in the 12 months before the start of the index spell, due to stress-related disorders (ICD-10 F43), due to other mental disorders (ICD-10 F00-F99 and Z73, excluding F43), or due to any somatic disorders (excluding ICD-10 A00-Z99, excluding F00-F99, O80 and Z00-Z99): (0, ≥1 inpatient days and 0, 1, 2–3, or >4 outpatient visits).

## Statistical analysis

Statistical analysis was conducted separately for each cohort. Group-based trajectory modeling was used to identify latent groups of individuals who had similar trajectories of average monthly SA/DP days in the 13 months following a new SA spell due to stress-related disorders. Several potential models were estimated with between two and nine trajectory groups with varying polynomial shapes (linear, quadratic, and cubic). In order to facilitate model convergence, intercept models were first fit to the data, and the results of these were used as starting values when estimating more complex trajectory shapes. Several metrics were considered when determining the optimal number of trajectories, including the Bayesian Information Criterion (BIC), Akaike information criterion (AIC), the average posterior probability of assignment (APPA), the entropy of the posterior probabilities, the root mean squared error (RMSE), and the odds of correct classification (OCC). In order to keep the number of groups within reach for interpretation, a requirement of a minimum of 5% of the study population for the smallest group was introduced [27]. Finally, individuals were assigned trajectory group membership based on their posterior assignment probability.

To explore the characteristics of the identified trajectories, we first presented the distribution of sociodemographic, work-, and morbidity-related variables by trajectory group memberships. Then, we conducted a multinomial logistic regression to examine the associations between each covariate and trajectory group membership while mutually adjusting for all other covariates. We used the trajectory group with most individuals as the reference group and presented mutually adjusted odds ratios (OR) with 95% confidence intervals (CI) for each covariate. Besides calculating ORs, we also estimated the strength of associations between each covariate and trajectory membership by comparing the fully adjusted model with reduced models by excluding one covariate at a time, performing likelihood ratio tests, and computing the differences in Nagelkerke pseudo $R^2$ between the full and reduced models. We also present effect displays [28], to show the estimated probability of trajectory group membership across the different covariate categories. The trajectory analysis was performed in SAS (version 9.4) using the procedure 'Proc Traj'. The remaining statistical analyses were performed in R software version 4.3.1 using the *nnet*, *lmtest*, *DescTools*, and *effects* packages. Covariate categories with an overall frequency of less than 0.5% were combined with other categories. In a sensitivity analysis, individuals in the datasets who had missing values on any of the covariates were excluded. Otherwise, missing values were treated as a separate category in the analysis. The project was approved by the Regional Ethical Review Board of Stockholm, Sweden (2007/5:6) and the Swedish Ethical Review Authority (Dnr. 2021-06441-02), who also waived the need for informed consent.

## Results and discussion

### Results

The distribution of sociodemographics, work-, and morbidity-related variables of each cohort is presented in Table 1. In the Cohort-2011 (n = 32,417) 76.5% were women, 37.1% were 40 years or younger, and 86.6% were born in Sweden. Among the 65,511 individuals with a new SA spell >14 days in 2018 (cohort-2018), 76.1% were women, 43.1% were 40 years or younger, and 85.5% were born in Sweden. Most individuals had more than the compulsory educational level (90.5% and 92.4% in Cohort-2011 and Cohort-2018, respectively). Most were in paid work when the SA spell started (92.8% and 95.9%), and for most, the SA started on full-time (80.0%, 81.6%). During the year before the index spell, the majority had not received SA benefits, nor been hospitalized—however, nearly half had one or more specialised outpatient-care visit. A small percentage of people had specialised outpatient care visits due to F43 the year before the start of index spell (3.5% in both cohorts), or for other mental diagnosis (5.6%, 6.3%). The mean of SA/DP days was 101 (Standard deviation (SD): 115) in the 2011 cohort and 106 (SD: 106) in the 2018 cohort during the 13-month follow-up.

### Trajectory analyses

A six-group solution for both cohorts was selected (Fig 1). The final models had APPA values of 0.95 and OCC values above 24 in all trajectory groups, which indicates good model fits. The shapes of the trajectories were similar in the two cohorts, though the size of the group memberships differed. The trajectories were labeled as follows: *steep drop* (35.9% of all in the Cohort-2011 and 30.6% in the Cohort-2018); *constant fluctuating* (11.2%, 8.7%), *fast decrease* (24.4%, 25.5%); *medium decrease* (13.1%, 18.1%), *slow decrease* (7.3%, 10.8%), and *constant high* (8.0%, 6.2%).

Four of the trajectory groups (*steep drop*, *fast*, *medium decrease*, *slow decrease*) had zero or close to zero average net days of SA/DP during the follow-up. This corresponded to almost two-thirds of the cohort. More than 30% of the cohorts were classified to a trajectory that

**Table 1. Study population and background characteristics among individuals with an incident sickness absence (SA) spell due to stress-related disorders in 2011 and in 2018, respectively.**

| | Cohort-2011 | Cohort-2018 |
|---|---|---|
| | (N = 32,417) | (N = 65,511) |
| **Sex** | | |
| Women | 24,810 (76.5%) | 49,825 (76.1%) |
| Men | 7607 (23.5%) | 15,686 (23.9%) |
| **Age** | | |
| 18–30 | 3711 (11.4%) | 10,732 (16.4%) |
| 31–40 | 8321 (25.7%) | 17,505 (26.7%) |
| 41–50 | 10,156 (31.3%) | 18,467 (28.2%) |
| 51–64 | 10,229 (31.6%) | 18,807 (28.7%) |
| **Country of birth** | | |
| Sweden | 28,061 (86.6%) | 55,984 (85.5%) |
| Other nordic countries | 956 (2.9%) | 1293 (2.0%) |
| Other EU25 countries | 682 (2.1%) | 1598 (2.4%) |
| Rest of the world | 2718 (8.4%) | 6636 (10.1%) |
| **Level of education** | | |
| Primary (≤9 years) | 3089 (9.5%) | 4984 (7.6%) |
| High school (10–12 years) | 14,928 (46.1%) | 28,846 (44.0%) |
| College, university (>12 years) | 14,400 (44.4%) | 31,681 (48.4%) |
| **Type of living area** | | |
| Big city | 13,327 (41.1%) | 23,700 (36.2%) |
| Medium-sized city | 10,862 (33.5%) | 25,916 (39.6%) |
| Small city/village | 8228 (25.4%) | 15,895 (24.3%) |
| **Family situation** | | |
| Married/cohabitant without children | 4531 (14.0%) | 11,214 (17.1%) |
| Married/cohabitant with children | 13,582 (41.9%) | 23,243 (35.5%) |
| Single without children | 9746 (30.1%) | 24,368 (37.2%) |
| Single with children | 4558 (14.1%) | 6686 (10.2%) |
| **Occupational status** | | |
| White collar | 17,178 (53.0%) | 34,232 (52.3%) |
| Blue collar | 14,075 (43.4%) | 25,900 (39.5%) |
| Missing information | 1164 (3.6%) | 5379 (8.2%) |
| **Employment status at the start of the sickness absence spell** | | |
| Employed | 30,067 (92.8%) | 62,796 (95.9%) |
| Unemployed | 1984 (6.1%) | 2071 (3.2%) |
| Parental leave/homemaker | 346 (1.1%) | 183 (0.3%) |
| Student | 20 (0.1%) | 9 (0.0%) |
| Missing | | 452 (0.7%) |
| **Extent at the start of the sickness absence spell** | | |
| 25% | 1226 (3.8%) | 2115 (3.2%) |
| 50% | 4433 (13.7%) | 8475 (12.9%) |
| 75% | 826 (2.5%) | 1445 (2.2%) |
| 100% | 25,932 (80.0%) | 53,476 (81.6%) |
| **In the previous 12 months** | | |
| **Sickness absence days due to stress-related diagnosis** | | |
| 0 | 30,886 (95.3%) | 62,141 (94.9%) |
| 0.25–50 | 907 (2.8%) | 1896 (2.9%) |

(*Continued*)

**Table 1.** (Continued)

| | Cohort-2011 | Cohort-2018 |
|---|---|---|
| | **(N = 32,417)** | **(N = 65,511)** |
| 50.25–90 | 241 (0.7%) | 621 (0.9%) |
| 90.25–180 | 255 (0.8%) | 629 (1.0%) |
| 180.25- | 128 (0.4%) | 224 (0.3%) |
| **Sickness absence days due to other mental diagnosis** | | |
| 0 | 31,192 (96.2%) | 63,759 (97.3%) |
| 0.25–50 | 698 (2.2%) | 997 (1.5%) |
| 50.25–90 | 201 (0.6%) | 325 (0.5%) |
| 90.25–180 | 182 (0.6%) | 287 (0.4%) |
| 180.25- | 144 (0.4%) | 143 (0.2%) |
| **Sickness absence days due to somatic diagnosis** | | |
| 0 | 28,710 (88.6%) | 62,162 (94.9%) |
| 0.25–50 | 2939 (9.1%) | 2512 (3.8%) |
| 50.25–90 | 381 (1.2%) | 437 (0.7%) |
| 90.25–180 | 263 (0.8%) | 294 (0.4%) |
| 180.25- | 124 (0.4%) | 106 (0.2%) |
| **Disability pension** | | |
| None | 30,334 (93.6%) | 64,357 (98.2%) |
| Any | 2083 (6.4%) | 1154 (1.8%) |
| **Specialized outpatient healthcare visits** | | |
| 0 | 31,283 (96.5%) | 63,250 (96.5%) |
| 1 | 622 (1.9%) | 1108 (1.7%) |
| 2–3 | 322 (1.0%) | 729 (1.1%) |
| ≥4 | 190 (0.6%) | 424 (0.6%) |
| **Specialized outpatient healthcare visits with other mental diagnosis** | | |
| 0 | 30,560 (94.3%) | 61,371 (93.7%) |
| 1 | 888 (2.7%) | 1967 (3.0%) |
| 2–3 | 565 (1.7%) | 1294 (2.0%) |
| ≥4 | 404 (1.2%) | 879 (1.3%) |
| **Specialized outpatient healthcare visits with somatic diagnosis** | | |
| 0 | 19,664 (60.7%) | 37,465 (57.2%) |
| 1 | 6170 (19.0%) | 12,500 (19.1%) |
| 2–3 | 4356 (13.4%) | 9736 (14.9%) |
| ≥4 | 2227 (6.9%) | 5810 (8.9%) |
| **Inpatient healthcare days with stress-related disorder** | | |
| No | 32,197 (99.3%) | 65,206 (99.5%) |
| Yes | 220 (0.7%) | 305 (0.5%) |
| **Inpatient healthcare days with somatic diagnosis** | | |
| No | 29,834 (92.0%) | 61,653 (94.1%) |
| Yes | 2583 (8.0%) | 3858 (5.9%) |
| **Inpatient healthcare days for other mental diagnosis** | | |
| No | 32,030 (98.8%) | 64,877 (99.0%) |
| Yes | 387 (1.2%) | 634 (1.0%) |

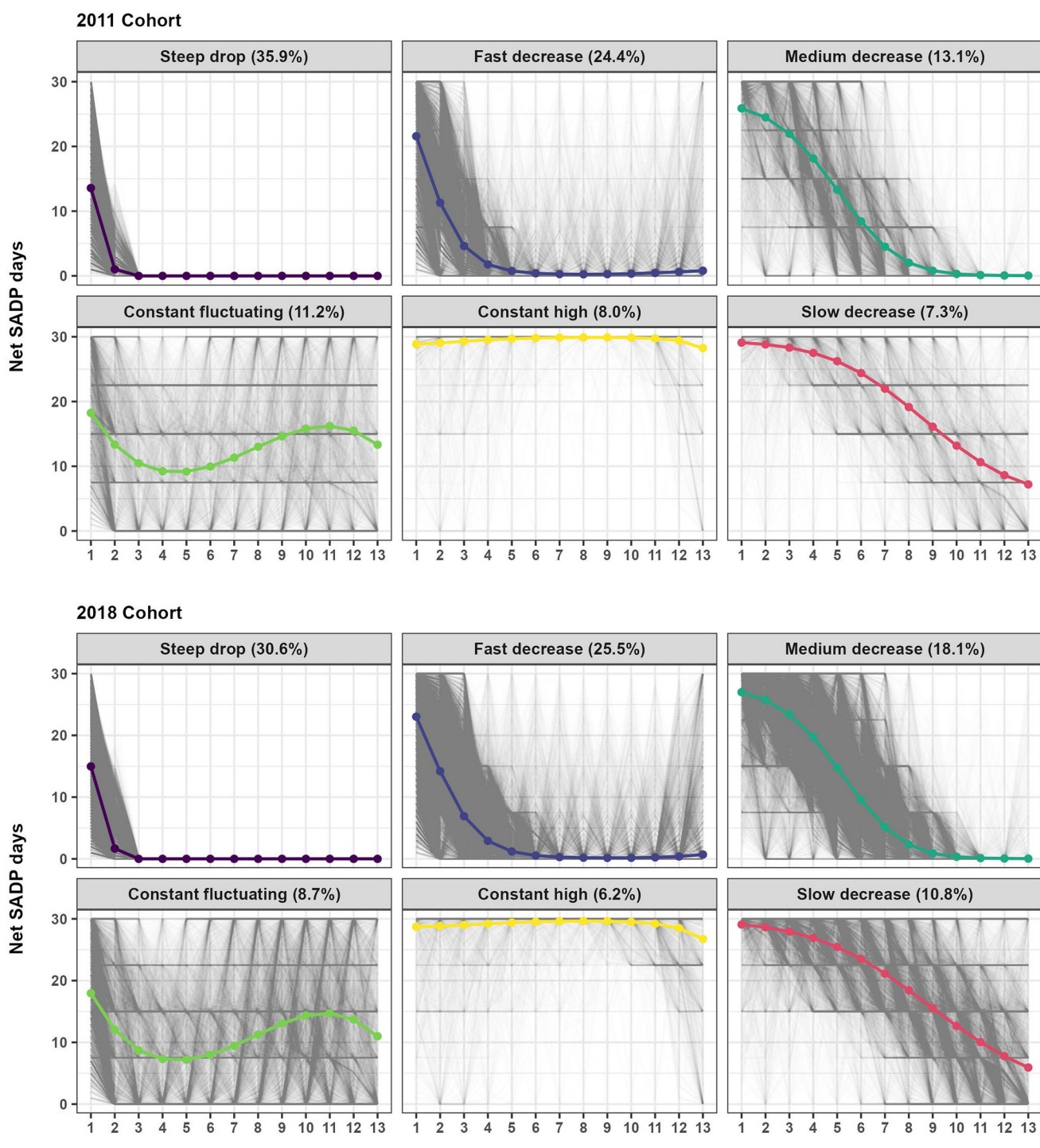

**Fig 1. Trajectories of sickness absence and disability pension days and individual trajectories amongst individuals with sickness absence due to stress-related disorders in 2011 and 2018.**

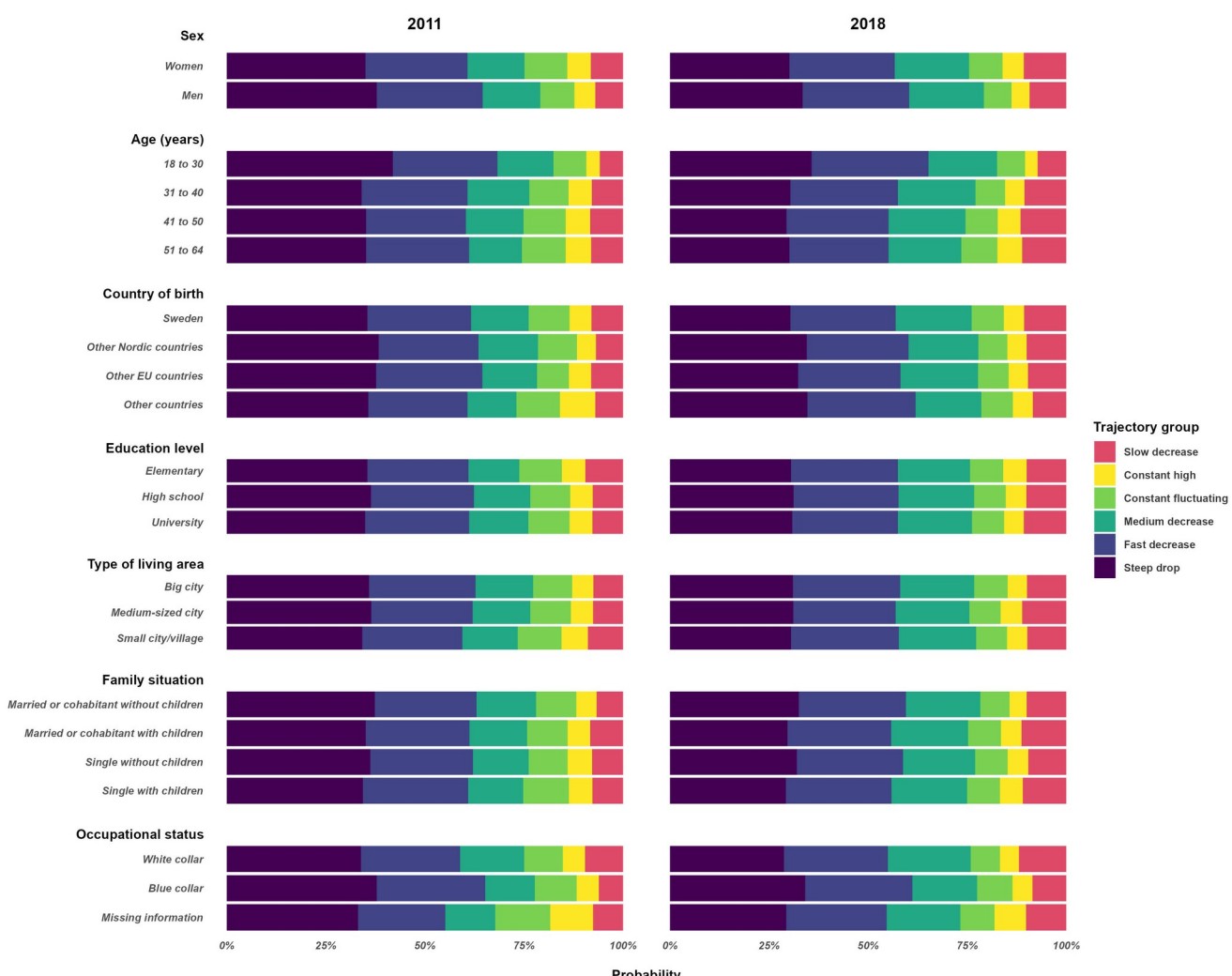

**Fig 2. Estimated probabilities of trajectory group membership across sociodemographic and socioeconomic covariates in Cohort-2011 and Cohort-2018.**

reached zero average net SA/DP days within three months (*steep drop* trajectory). The second largest trajectory (*fast decrease*) was approaching zero average net days of SA/DP after about seven months. Less than 10% of the cohorts were classified to the trajectory (*constant high)* which had a constant high average number of SA/DP days/month throughout the follow-up. Metrics of alternative trajectory models are presented in S3 Table.

### Distribution of sociodemographic, work-, and morbidity-related covariates

Descriptive statistics regarding the distributions of the sociodemographic, work, and morbidity-related covariates over the trajectories are presented in S1 Table. The effect displays of the estimated probability of trajectory group membership by sociodemographic and work- and morbidity-related covariates are presented in Figs 2–4.

Both in Cohort-2011 and in Cohort-2018, the *steep drop* trajectory had the highest percentage of men (25.5%, 26.6%, respectively), young individuals aged 18–30 (14.5%, 19.7%), no SA days due to stress-related (97.2%, 96.8%), other mental (97.9%, 98.4%), or somatic diagnosis

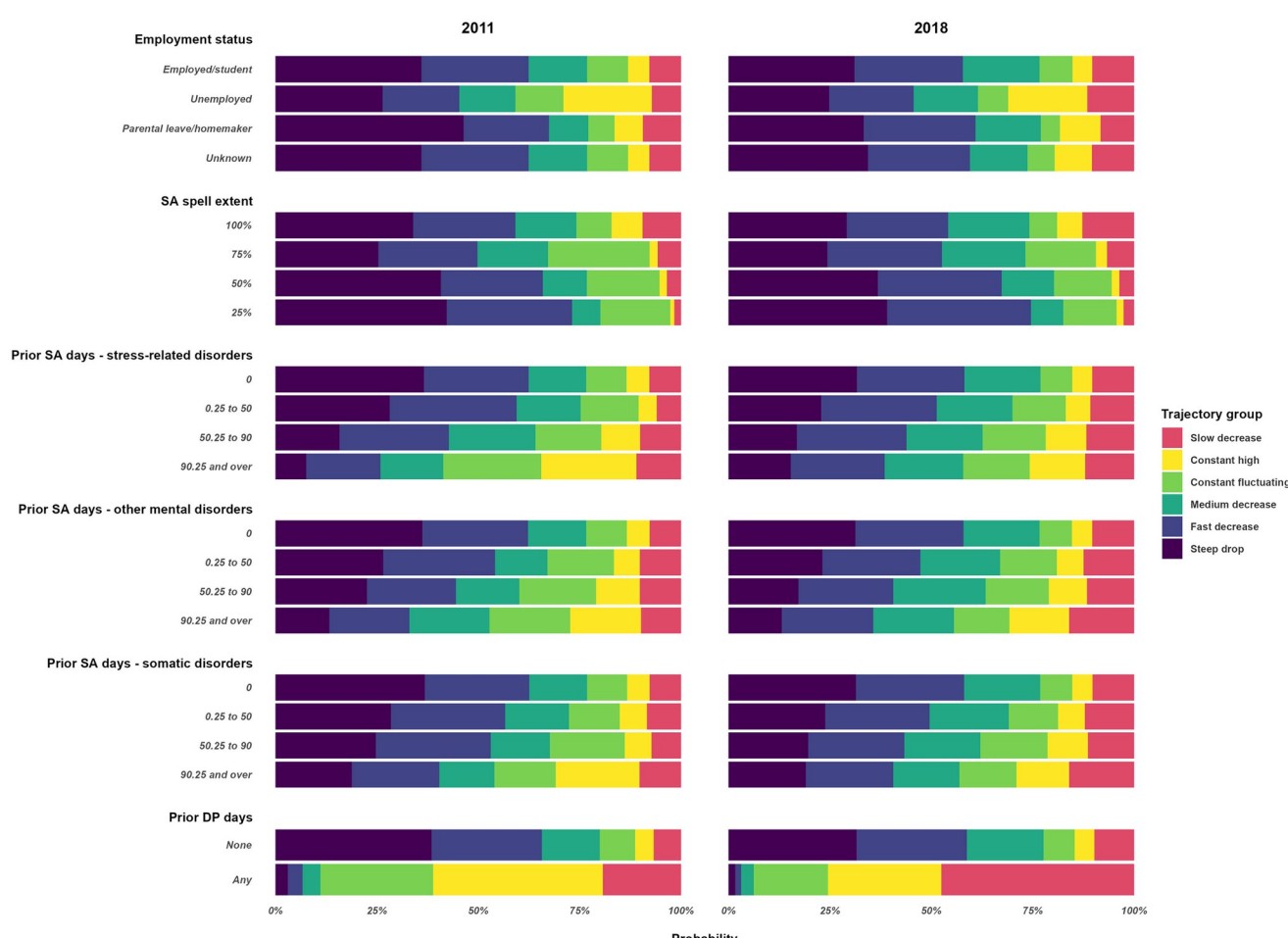

**Fig 3. Estimated probabilities of trajectory group membership across employment and sickness absence-related covariates in Cohort-2011 and Cohort-2018.**

(91.7%, 96.5%), and hardly any with DP (99.5%, 99.9%). The highest prevalence of women (82.9%, 80.7%), the lowest prevalence of full-time SA at the beginning of the spell (52.1%, 61.9%), and the highest prevalence of any SA days due to stress-related disorders one year before the index spell (7.7%, 10.3%) were in *the constant fluctuating* trajectory. The highest prevalence of individuals with ≤9 years of education (13.5%, 10.6%) and unemployment (31.6%, 15.7%) were in the *constant high* trajectory while the highest prevalence of white-collar workers was in the *slow decrease* trajectory (61.0%, 60.3%). The mutually adjusted associations between each covariate and trajectory memberships is presented in S2 Table. The forest plots by groups of covariates are presented in S1–S5 Figs. Compared to the *steep drop* trajectory, being a woman, having SA or healthcare use in the year before the index spell started was associated with belonging to all other trajectory groups. Having previous SA due to stress-related or other mental diagnoses was a strong independent predictor of belonging to the *constant fluctuating* and *constant high trajectory*. Older age was a stronger independent predictor of belonging to the *slow decrease* and *constant high* trajectory groups. White-collar work was associated with belonging to the *slow* (in both cohorts) *and medium decrease* trajectories, and being unemployed at the start of the SA spell was the strongest predictor of belonging to the

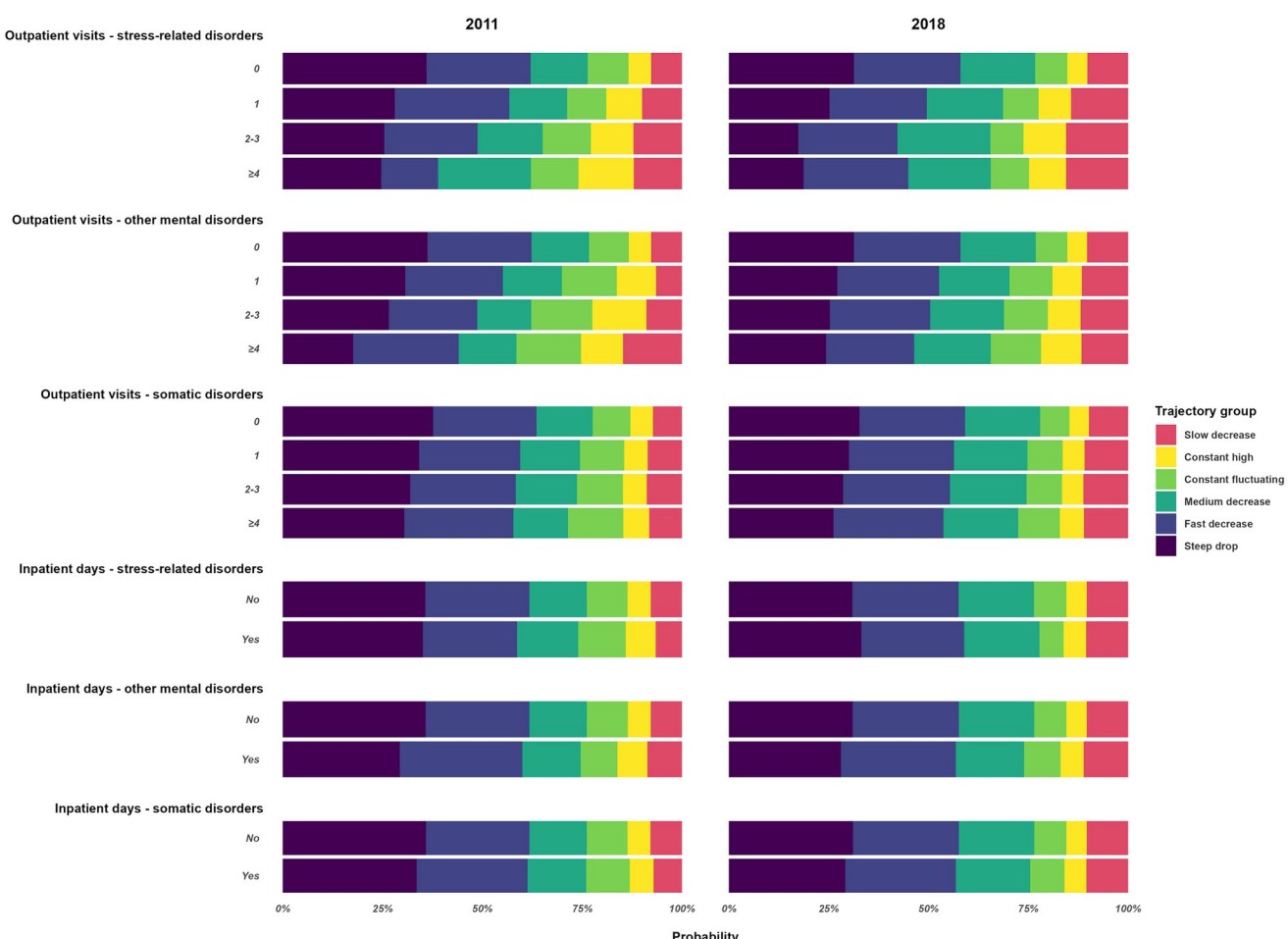

**Fig 4. Estimated probabilities of trajectory group membership across secondary healthcare use related covariates in Cohort-2011 and Cohort-2018.**

*constant high* trajectory. Starting the SA with full time was a strong predictor for belonging to the *slow decrease* and *constant high* trajectories. Having previous SA or healthcare visits due to stress-related and other mental disorders were strong predictors for the *constant fluctuating*, *constant high*, and *slow decrease* trajectories.

The difference in Pseudo $R^2$, AIC, and BIC between the fully adjusted model and the fully adjusted model without a given covariate is presented in S4 Table. The largest decreases in the $R^2$ were measured when removing SA extent at the start of the spell (3.2 in Cohort-2011 and 3.5 in Cohort-2018) and prior DP (6.8 in Cohort-2011 and 3.0 in Cohort-2018). A substantial decrease in the $R^2$ was observed when age, occupation, employment, and prior SA due to stress-related disorders were removed from the model in both cohorts and prior SA due to mental and somatic disorders in Cohort-2011.

## Discussion

In this exploratory prospective cohort study of all people in Sweden who, in 2011 or in 2018, had a new SA spell >14 days due to stress-related diagnoses, we identified six trajectories of future SA/DP net days in both cohorts during the 13-month follow-up. The largest trajectory group was the *steep drop* trajectory, characterised by a sharp drop to zero SA/DP days within 3

months and zero SA/DP during the rest of the follow-up. The vast majority of individuals in both cohorts belonged to trajectory groups that reached zero SA/DP at the end of the follow-up, but we also identified trajectories that indicated long-term SA/DP without reaching zero days during the follow-up. Sociodemographics, work, and morbidity-related characteristics differed by trajectory groups. Being a woman, of older age, being unemployed at the start of the SA, having prior SA/DP, and prior secondary healthcare use were predictors of belonging to either the *constant high*, the *constant fluctuating*, or the *slow decrease* trajectories, trajectories that indicated slow or no return-to-work patterns. While Cohort-2018 was double the size compared to Cohort-2011, the identified trajectory groups, the distributions of individuals by trajectory groups, and the major predictors associated with the trajectories were similar in the two cohorts.

Our findings suggest that individuals follow different SA/DP trajectories when starting a SA spell due to stress-related disorders. While the incidence of SA due to stress-related disorders has almost doubled between 2011 and 2018, the trajectory groups we identified in the cohorts were similar, and the vast majority reached zero SA days during the 13-month follow-up. Only the *constant high trajectory*, in which individuals followed SA/DP trajectories with stable, high level of monthly SA/DP days, and the *constant fluctuating trajectory*, in which individuals still had SA/DP days at the end of the follow-up and thus had trajectories that indicate no or partial return to previous activities. The largest trajectory was the *steep drop* trajectory in both cohorts, which could be described as a rapid drop to zero SA days within approximately three months after the SA spell began. This trajectory group was associated with characteristics such as no SA, DP, or secondary healthcare use in the prior year, male sex, and younger age, which is consistent with findings from studies among individuals on SA due to common mental disorders (including anxiety, mild-, moderate depression, and stress-related disorder), which found that the majority of people on SA who have no multi-morbidities nor previous SA recover relatively fast and can return to work [20]. Female sex, older age, previous SA, unemployment, and multimorbidity were identified as major predictors of more SA days three years after the index SA spell [21]. On the other hand, the *constant fluctuating* and the *constant high* trajectories, together holding 20% and 16% of the individuals in 2011 and 2018, indicate long-term SA and DP patterns. Individuals in these trajectory groups were more likely to have prior SA and secondary healthcare use due to stress-related and other mental diagnoses, were more likely on partial DP at the start of the index spell, and were more likely to be older and a woman. Similar risk factors were identified in previous studies in relation to long-term SA, DP, and lower return-to-work rate among individuals on SA due to stress-related and common mental disorders [18,19,21]. Studies among individuals on SA due to mental disorders, but not specifically stress-related or common mental disorders, reported that older age is consistently associated with negative work-related outcomes, such as longer time to return to work and DP, but other factors such as multi-morbidity, male sex, medium/higher education, unemployment, and continuing disability were also associated with longer time to return to work [13]. We found that being unemployed at the start of the SA spell was a strong predictor for belonging to the *constant high* trajectory group, which might suggest that these individuals were already in a marginalized position in the labor market. The strong association between morbidity-related predictors with belonging to this trajectory group also supports that this group might have difficulties staying attached to the labor market due to morbidity-related factors. However, unemployment was not a strong predictor for other less favorable trajectories, such as the *constant fluctuating* or *slow decrease* trajectories. Individuals in these groups might still be attached to the labor market, in spite of previous morbidity-related histories. Starting the SA spell on full-time was an independent strong predictor of belonging to the *constant high* and the *slow decrease* trajectories, while starting SA on part-time was negatively associated

with belonging to the *constant fluctuating* trajectory. This suggests that the *constant fluctuating* trajectory might include individuals whose work incapacity was not as severe why they were able to work some, and did not need full-time SA/DP [20].

Notably, the two cohorts were similar concerning the identified trajectories, the size of the trajectory groups, and the associated predictors, even though the 2018 cohort was almost double the size of the 2011 cohort. The increase in the incidence in SA spells due to stress-related disorders might partly be due to increase of "exhaustion disorder" diagnoses within the F43 diagnostic group [4]. The "exhaustion disorder" diagnosis was introduced in the Swedish ICD-10th version in 2005, and while it has never been validated, it is conceptually similar to the burnout construct which has been recognized worldwide and considered a clinical manifestation of burnout [8]. As reported by the latest statistics, the exhaustion disorder diagnosis contributes to more than half of the SA spells >14 days within the F43 group [4]. The prevalence of psychological distress and individuals who report stress-related symptoms has increased substantially during the period [29], which might explain part of the increase in stress-related SA incidence. It is also likely that with the introduction of the ICD code for exhaustion disorder, the diagnosis might capture more individuals who previously were diagnosed either with burn-out depression, other somatoform disorders, or related insomnia [30]. However, the trajectories and the risk factors for the different trajectories seem to stay relatively stable, indicating similar subgroup structures within this diagnostic group.

## Strengths and limitations

This is the first large prospective cohort study regarding trajectories of future work SA/DP days among all people with a new SA spell >14 days due to stress-related disorders (F43). One of the strengths are that we could use microdata linked from several nationwide administrative registers of good quality [23,24,31], covering the whole working-age population in Sweden. Other strengths are that both sociodemographic and morbidity information could be included, regarding specialised out- and inpatient healthcare. All such diagnoses were set by physicians and classified according to ICD-10. An additional strength is that we could include both SA and DP net days during the follow-up, which gives a better indication of future work participation compared to measuring only SA or DP separately. Furthermore, we included a recent and an older cohort, which made it possible to examine possible changes in trajectories and the in the distribution of predictors by the identified trajectory at the beginning of the strong positive trend in incident SA due to stress-related disorder and in a more recent period just before the Covid-19 pandemic.

There were also limitations to this study. While the SA diagnosis of stress-related disorder was set by the physician, and the validity of SA diagnoses at a three-digit level has been found to be high when compared with diagnoses in medical files [32], we cannot rule out that some of the individuals were misdiagnosed as symptoms of stress-related disorders might overlap with other mental or somatic diseases [33,34]. As awareness and acceptance of common mental health disorders, especially stress-related disorder, has increased substantially during the last decades, it is hypothesised that this can contribute to a more accurate and earlier symptom detection [35]. Moreover, we did not have information on primary healthcare. Another limitation of our study is that as the first 14 days of sickness absence is mostly administered by employers in Sweden, our study did not include those individuals with less severe symptoms of stress-related disorders who only take short sickness absence spells. Furthermore, while we examined a wide range of sociodemographic and health-related possible predictors in relation to future SA/DP trajectories, other factors such as lifestyle, type of rehabilitation measures, perceived job strain and social support, work environment, and work contract or the economy of

the company might also be associated with the observed trajectories. Finally, we only had information about the main SA diagnoses in the first SA period of the first SA spell during the follow-up period, i.e., change of diagnosis in a SA spell was not recorded. The differences in social welfare systems and in diagnostic systems limit comparability with findings in other countries. While the strict external validity of our findings is limited to high-income countries with similar social welfare and diagnostic systems, it is likely that similar groups of SA/DP trajectories can be identified in other countries in similar patient groups.

## Conclusions

In this exploratory study, we identified six different future SA/DP day trajectories within two cohorts of individuals who during 2011 and 2018, respectively, had a new SA spell >14 days due to stress-related disorders. While the largest trajectory included individuals, who reached zero SA/DP after a couple of months from the beginning of the SA spell and the majority of individuals reached near zero SA by the end of the follow-up, some individuals followed less favourable SA/DP trajectories. The trajectories differed by their associations of socioeconomic, morbidity-, and work-related predictors, nevertheless, were similar in the two cohorts. Being a woman, older, and having prior SA or secondary healthcare use due to mental disorders were associated with the *constant high*, *constant fluctuating*, and *slow decrease* trajectories, suggesting needs of targeted intervention strategies, and monitoring for these subgroups to avoid adverse labor market outcomes.

## Supporting information

**S1 Fig. Odds ratios and 95% confidence intervals for the associations between sociodemographic and occupation-related variables and trajectory memberships in Cohort-2011 and Cohort-2018 with the "steep drop" trajectory as reference.**
(TIF)

**S2 Fig. Odds ratios and 95% confidence intervals for the associations between sickness absence (SA) histories and trajectory memberships in Cohort-2011 and Cohort-2018 with the "steep drop" trajectory as reference.**
(JPEG)

**S3 Fig. Odds ratios and 95% confidence intervals for the associations between employment, extent of sickness absence (SA), disability pension at the start of the SA spell, and trajectory memberships in Cohort-2011 and Cohort-2018 with the "steep drop" trajectory as reference.**
(TIF)

**S4 Fig. Odds ratios and 95% confidence intervals for the associations between specialized healthcare use and trajectory memberships in Cohort-2011 and Cohort-2018 with the "steep drop" trajectory as reference.**
(TIF)

**S5 Fig. Odds ratios and 95% confidence intervals for the associations between inpatient healthcare use and trajectory memberships in Cohort-2011 and Cohort-2018 with the "steep drop" trajectory as reference.**
(TIF)

**S1 Table. Distributions of sociodemographic characteristics in each trajectory group of monthly sickness absence and disability pension days among the individuals with a new**

sickness absence spell due to stress-related disorders.
(DOCX)

**S2 Table. Distributions of sociodemographic characteristics in each trajectory group of monthly sickness absence and disability pension days among the individuals with a new sickness absence spell due to stress-related disorders in 2011 and in 2018.**
(DOCX)

**S3 Table. Statistical characteristics of alternative trajectory models for the 2011 and 2018 cohort.**
(DOCX)

**S4 Table. The goodness of fit measures, comparing multinomial logistic regressions (fully adjusted and with one covariate excluded at a time) modeling the associations between sociodemographic, work, and health-related variables with trajectory group members, amongst individuals with sickness absence due to stress-related disorders in 2011 and in 2018.**
(DOCX)

## Author Contributions

**Conceptualization:** Katalin Gémes, Sara Sjölund Andoff, Kristin Farrants, Emilie Friberg.

**Data curation:** Emma Pettersson.

**Investigation:** Katalin Gémes, Emma Pettersson.

**Methodology:** Katalin Gémes, Emma Pettersson, Sara Sjölund Andoff, Kristin Farrants, Emilie Friberg, Kristina Alexanderson.

**Project administration:** Kristina Alexanderson.

**Resources:** Kristina Alexanderson.

**Supervision:** Katalin Gémes, Kristina Alexanderson.

**Visualization:** Emma Pettersson.

**Writing – original draft:** Katalin Gémes, Sara Sjölund Andoff.

**Writing – review & editing:** Katalin Gémes, Emma Pettersson, Sara Sjölund Andoff, Kristin Farrants, Emilie Friberg, Kristina Alexanderson.

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
