## [Decision Letter · Decision Letter 0]

8 Oct 2024

PONE-D-24-27390Sickness absence and disability pension trajectories among individuals on sickeness absence due to stress-related disorders. Two prospective population-based cohorts with 13-month follow-upPLOS ONE

Dear Dr. Gémes,

Thank you for submitting your manuscript to PLOS ONE. After careful consideration, we feel that it has merit but does not fully meet PLOS ONE’s publication criteria as it currently stands. Therefore, we invite you to submit a revised version of the manuscript that addresses the points raised during the review process.

You find our reviewer comments below.

We look forward to receiving your revised manuscript.

Kind regards,

Thomas Behrens

Academic Editor

PLOS ONE

“This study was financially supported by the Research Council for Health, Working Life and Welfare and the Swedish Social Insurance Agency. We utilised data from the REWHARD consortium supported by the Swedish Research Council (grant no. 2021-00154).”

“We utilised data from the REWHARD consortium supported by the Swedish Research Council (grant no. 2021-00154).”

“This study was financially supported by the Research Council for Health, Working Life and Welfare and the Swedish Social Insurance Agency. We utilised data from the REWHARD consortium supported by the Swedish Research Council (grant no. 2021-00154).”

“None”

5. In the online submission form you indicate that your data is not available for proprietary reasons and have provided a contact point for accessing this data. Please note that your current contact point is a co-author on this manuscript. According to our Data Policy, the contact point must not be an author on the manuscript and must be an institutional contact, ideally not an individual. Please revise your data statement to a non-author institutional point of contact, such as a data access or ethics committee, and send this to us via return email. Please also include contact information for the third party organization, and please include the full citation of where the data can be found.

Reviewers' comments:

Reviewer's Responses to Questions

**Comments to the Author**

1. Is the manuscript technically sound, and do the data support the conclusions?

Reviewer #1: Yes

Reviewer #2: Yes

2. Has the statistical analysis been performed appropriately and rigorously? 

Reviewer #1: Yes

Reviewer #2: Yes

3. Have the authors made all data underlying the findings in their manuscript fully available?

Reviewer #1: No

Reviewer #2: No

4. Is the manuscript presented in an intelligible fashion and written in standard English?

Reviewer #1: Yes

Reviewer #2: Yes

5. Review Comments to the Author

Reviewer #1: Comments

1. The analysis does not include data from Swedish primary healthcare. This could be significant since primary care is often the first point of contact for patients with stress-related disorders also in Sweden.

2. The study relies on diagnoses set by Swedish physicians for stress-related disorders, but it acknowledges the possibility of misdiagnosis. The overlap in symptoms between stress-related disorders and other mental or somatic diseases could lead to inaccuracies in the classification, potentially biasing the estimates that are presented in the paper.

3. The analysis only considers the main diagnosis at the start of the sickness absence spell. This implies that changes in diagnosis during the follow-up period are not accounted for in the analysis. This overlooks any shifts in the health status of individuals, affecting the interpretation of the results that are presented in the paper.

4. The study focuses on sickness absence spells longer than 14 days, excluding individuals with shorter-term absences who may have different characteristics and recovery trajectories. This could lead to biased estimates of the population’s overall stress-related disorder burden in Sweden.

5. Although the study includes some work-related variables, it does not account for specific occupational factors such as job stress levels, work environment, or employer support. These most likely have substantial influence the duration and recurrence of sickness absence.

6. The follow-up period of 13 months may not be sufficient to capture long-term trajectories of sickness absence and disability pension, especially for chronic stress-related conditions.

7. The trajectory analysis relies on several model specifications and technical assumptions (e.g., choice of the number of trajectory groups, polynomial shapes). The results could be sensitive to these choices, raising some concerns about the robustness of the empirical findings that are presented in the paper.

8. There are earlier empirical studies relevant to this topic from other Nordic countries that are not covered in this paper (see https://doi.org/10.1136/oemed-2020-106660).

9. What is the external validity of the estimation results that are presented in the paper for other high-income countries than Sweden?

10. Are there practical policy implications for other countries than Sweden?

Reviewer #2: Gémes et al. have analysed trajectories of sickness absence and disability pension following an initial sickness absence due to stress-related mental disorders. Overall, the manuscript is well written and well structured. Some questions / comments may be considered:

1. Was the outcome disability pension specific for stress-related disorders or general, due to all diseases? The outcomes (SA and DP) are missing in table1.

2. In the methods, authors describe how they handled missing values in general. Is there a reason to combine missings with one category in the education variable? Maybe, missing education could at least be included in table 1.

3. As the trajectories for both cohorts look very similar, were they calculated completely separately?

4. Many results are presented, partially redundant. As a suggestion, figures 2-4 (on p.8 accidentally “Figures 1, 2 ,3”?) could be placed in the supplement. If I understood correctly, trajectories were assigned to different degrees of severity of SA/DP, this could be used for ordering the trajectories in the figures rather than alphabetical ordering.

There is a typo in the title (‘sickeness’).

6. PLOS authors have the option to publish the peer review history of their article (what does this mean?). If published, this will include your full peer review and any attached files.

Reviewer #1: No

Reviewer #2: No

---

## [Author Response · Author response to Decision Letter 0]

12 Nov 2024

Manuscript number: PONE-D-24-27390

Manuscript title: Sickness absence and disability pension trajectories among individuals on sickness absence due to stress-related disorders. Two prospective population-based cohorts with 13-month follow-up

We appreciate the editors' and reviewers' constructive comments. Below, we address each point raised and explain how the manuscript has been revised in response to each. 

EDITOR

Comment

“1. Please ensure that your manuscript meets PLOS ONE's style requirements, including those for file naming. The PLOS ONE style templates can be found at

”

Authors’ response

The manuscript has been formatted according to PlosOne’s requirements.

Comment

“2. Thank you for stating the following financial disclosure:

“This study was financially supported by the Research Council for Health, Working Life and Welfare and the Swedish Social Insurance Agency. We utilised data from the REWHARD consortium supported by the Swedish Research Council (grant no. 2021-00154).”

Please include this amended Role of Funder statement in your cover letter; we will change the online submission form on your behalf.”

Authors’ response

The funders had no role in study design, data collection and analysis, decision to publish, or preparation of the manuscript. We have now included a statement about this to the updated cover letter. 

Comment

“We utilised data from the REWHARD consortium supported by the Swedish Research Council (grant no. 2021-00154).”

“This study was financially supported by the Research Council for Health, Working Life and Welfare and the Swedish Social Insurance Agency. We utilised data from the REWHARD consortium supported by the Swedish Research Council (grant no. 2021-00154).”

Authors’ response

We have now removed the funding statement from the manuscript. As the two statement are essentially the same, there is no need to modify the Funding statement.

Comment

“None”

Authors’ response

None of the authors have competing interests and we now we have added the requested statement to the updated cover letter.

Comment

5. In the online submission form you indicate that your data is not available for proprietary reasons and have provided a contact point for accessing this data. Please note that your current contact point is a co-author on this manuscript. According to our Data Policy, the contact point must not be an author on the manuscript and must be an institutional contact, ideally not an individual. Please revise your data statement to a non-author institutional point of contact, such as a data access or ethics committee, and send this to us via return email. Please also include contact information for the third party organization, and please include the full citation of where the data can be found.

Authors’ response

We have now revised our data statement so that it includes a non-author institutional point of contact via the project functional e-mail-address.

Comment

“6. Please include your full ethics statement in the ‘Methods’ section of your manuscript file. In your statement, please include the full name of the IRB or ethics committee who approved or waived your study, as well as whether or not you obtained informed written or verbal consent. If consent was waived for your study, please include this information in your statement as well.”

Authors’ response

We have now included the following ethics statement in the “Methods” section in the revised manuscript file: “The project was approved by the Regional Ethical Review Board of Stockholm, Sweden (2007/5:6) and the Swedish Ethical Review Authority (Dnr. 2021-06441-02), who also waived the need for informed consent.”

Comment

“7. Please include captions for your Supporting Information files at the end of your manuscript, and update any in-text citations to match accordingly. Please see our Supporting Information guidelines for more information: .”

Authors’ response

Now we edited the Supporting Information files according to Plos One’s guidelines about this.

Comment

Authors’ response:

We have again checked the references. We have not cited any paper that later was retracted. Also, we have made no changes to the reference list.

REVIEWER 1

Comment

“1. The analysis does not include data from Swedish primary healthcare. This could be significant since primary care is often the first point of contact for patients with stress-related disorders also in Sweden.”

Authors’ response

We agree and are aware that having access to primary healthcare data would have enhanced our ability to capture prior diagnoses of stress-related disorders, a diagnosis often treated in primary healthcare settings, as primary healthcare is to be the first-line treatment for common mental disorders. Unfortunately, we did not have such data; they were not collected by the Swedish Board of Welfare. On the other hand, we had information about all sickness absence spells due to stress-related disorders (F43), irrespective of whether the sickness absence benefit was based on a sickness certificate initiated by a physician in primary healthcare or in secondary healthcare. That we did not have information in primary healthcare visits is now acknowledged as a limitation, in the Strengths and Limitations section.

Comment

“2. The study relies on diagnoses set by Swedish physicians for stress-related disorders, but it acknowledges the possibility of misdiagnosis. The overlap in symptoms between stress-related disorders and other mental or somatic diseases could lead to inaccuracies in the classification, potentially biasing the estimates that are presented in the paper.”

Authors’ response

We agree with your comment. However, the aim of this study was to explore different trajectories after a new sickness absence spells >14 days due to F43 – irrespective of how correct the F43 diagnosis at the sickness certificate was. The information in the sickness certificate is what the treating physician (who often put the diagnosis there) and the social insurance officer have to act on.

The validity of sick-leave diagnoses is often discussed, nevertheless, only a few studies have investigated this. We acknowledge that misdiagnosis is always a possibility for any medical condition and maybe especially so for so mental disorders. However, assessing the accuracy of medical diagnoses set by physicians falls outside the scope of this study. We included secondary healthcare use for other mental and somatic diagnoses among the possible predictors as an indicator of the mental and somatic health burden that might influence sickness absence trajectories.

Comment

“3. The analysis only considers the main diagnosis at the start of the sickness absence spell. This implies that changes in diagnosis during the follow-up period are not accounted for in the analysis. This overlooks any shifts in the health status of individuals, affecting the interpretation of the results that are presented in the paper.”

Author’s response

You are right in that many aspects, at different structural levels, during follow up might influence future number of sickness absence and disability pension days. However, the aim of this study was to investigate the number of sickness absence and disability pension days throughout the follow-up period, among individuals with a sickness absence spell that initially was due to F43, to identify individuals who follow similar trajectories/patterns regarding the number of such days during follow-up - and to investigate possible associations between the baseline sociodemographic and health-related factors and the identified trajectories. This approach allowed us to identify specific subgroups, for example, individuals who might be at risk of labor market marginalization and no return to work. The changes in the sickness absence diagnosis during the follow-up time were thus not a part of our study question and outside the scope of this study. Thank you for suggesting other studies with other aims, e.g., to also account for changes over time regarding sickness absence diagnoses and shifts in health status during follow-ups. That would involve using other type of data, another study design, and other analytical approaches than used in this study. We hope that our results will inspire future such studies.

Comment

“4. The study focuses on sickness absence spells longer than 14 days, excluding individuals with shorter-term absences who may have different characteristics and recovery trajectories. This could lead to biased estimates of the population’s overall stress-related disorder burden in Sweden.”

Author’s response

We did not aim to give any estimates of the population’s overall stress-related disorder burden in Sweden. To do that, other types of data would be needed – such as self-reported data and healthcare data from both primary and secondary healthcare. To some extent, also shorter sickness absence spells (as you suggest) as well as disability pension due to these diagnoses might be useful. 

However, for most diagnoses, the use of sickness absence data is not a good way to estimate occurrence of that diagnosis. Most people with different diagnoses are not on sickness absence due to them – as the diagnosis(es) do not affect that person’s function and work capacity (that is, the functions needed in his/her specific occupation) to such a high degree that sickness absence is required. Nevertheless, sickness absence can be seen as a good measure of the social consequences of a diagnosis, in terms not being able to work/be self-sufficient – provided that you are covered by the sickness absence benefit scheme.

Our aim was to explore the possibility among those who had a sickness absence spell due to F43 that had lasted for >14 days. That is information of interest for sickness absence certifying physicians, social insurance officers, employers, and others when managing possible prolongation of such cases.

Here, we wanted to use high-quality data on sickness absence for the whole of Sweden, which the MiDAS register kept by the Swedish Social Insurance Agency provides SA information from day 15 of sickness absence spells. The first 14 days of SA are paid and administered by the employees and, therefore, are not included in MiDAS. Moreover, the validity of the sickness absence diagnoses of the shorter sickness absence spells is low; for the first week, they are self-reported, and a physician’s medical certificate is not required until the 8th day of the spell.

To acknowledge this, we have now added text about this in the revised Strength and Limitation section: “Another limitation of our study is that as the first 14 days of sickness absence is mostly administered by employers in Sweden, our study did not include those individuals with less severe symptoms of stress-related disorders who only take short sickness absence spells”.

Comment

“5. Although the study includes some work-related variables, it does not account for specific occupational factors such as job stress levels, work environment, or employer support. These most likely have substantial influence the duration and recurrence of sickness absence.”

Author’s response

We agree that there are several other factors, at different structural levels in society, that might have influenced future trajectories of sickness absence and disability pension in this group. Both the aspects of work that you mention as well as lifestyle factors (tobacco, alcohol, drugs, exercise, sleep patterns, etc), how sex-segregated the work place and/or the occupation is, down-sizing/up-sizing of the company, economy developments, changes in rules, etcetera.. However, we did not have such information regarding our cohort members. Moreover, all did not have paid work; some were unemployed or students. Second, much of the information you mention would require survey data, which is a research method that is more prone to bias due to its own limitations, such as selective drop-out and measurement error due to self-assessment. We now acknowledge such limitations under the Strength and Limitation paragraph: “While we examined a wide range of sociodemographic and health-related possible predictors in relation to future SA/DP trajectories, other factors such as lifestyle, type of rehabilitation measures, perceived job strain and social support, work environment, work contract, economy of the company might also be associated with the observed trajectories .”

Comment

“6. The follow-up period of 13 months may not be sufficient to capture long-term trajectories of sickness absence and disability pension, especially for chronic stress-related conditions.”

Author’s response

We appreciate the reviewer's suggestion to investigate even more long-term sickness absence and disability pension trajectories among people sickness absent due to F43 – hopefully such studies will be conducted. However, for the purposes of this study, we focused on the one-year outcome (12+1-month follow-up). This is a common follow-up time in return-to-work studies, and in most countries, 12 months is the upper time limit for receiving sickness absence benefits. Our primary objective was to analyze this follow-up period at a population level and to map the diversity in sickness absence trajectories after a stress-related sickness absence diagnosis, as opposed to the commonly used dichotomous outcomes. Additionally, given the substantial increase in stress-related sickness absence incidence in Sweden since 2010, we wanted to determine how patterns of sickness absence following a stress-related diagnosis change over time. Therefore, we established two cohorts at the beginning and end of the 2010s, keeping in mind to avoid including follow-up periods that collide with the COVID-19 pandemic. This decision also contributed to keeping the follow-up period relatively short.

Comment

“7. The trajectory analysis relies on several model specifications and technical assum

---

## [Decision Letter · Decision Letter 1]

1 Dec 2024

Sickness absence and disability pension trajectories among individuals on sickness absence due to stress-related disorders. Two prospective population-based cohorts with 13-month follow-up

PONE-D-24-27390R1

Dear Dr. Gémes,

We’re pleased to inform you that your manuscript has been judged scientifically suitable for publication and will be formally accepted for publication once it meets all outstanding technical requirements.

Kind regards,

Thomas Behrens

Academic Editor

PLOS ONE

Reviewers' comments:

Reviewer's Responses to Questions

**Comments to the Author**

1. If the authors have adequately addressed your comments raised in a previous round of review and you feel that this manuscript is now acceptable for publication, you may indicate that here to bypass the “Comments to the Author” section, enter your conflict of interest statement in the “Confidential to Editor” section, and submit your "Accept" recommendation.

Reviewer #1: All comments have been addressed

Reviewer #2: All comments have been addressed

2. Is the manuscript technically sound, and do the data support the conclusions?

Reviewer #1: Yes

Reviewer #2: Yes

3. Has the statistical analysis been performed appropriately and rigorously? 

Reviewer #1: Yes

Reviewer #2: Yes

4. Have the authors made all data underlying the findings in their manuscript fully available?

Reviewer #1: No

Reviewer #2: No

5. Is the manuscript presented in an intelligible fashion and written in standard English?

Reviewer #1: Yes

Reviewer #2: Yes

6. Review Comments to the Author

Reviewer #1: I am happy with the revisions made in the paper and with the effort you put in replying to my comments.

Reviewer #2: (No Response)

7. PLOS authors have the option to publish the peer review history of their article (what does this mean?). If published, this will include your full peer review and any attached files.

Reviewer #1: **Yes: **Petri Böckerman

Reviewer #2: No

---

## [Editor Report · Acceptance letter]

4 Dec 2024

PONE-D-24-27390R1 

PLOS ONE

Dear Dr. Gémes, 

I'm pleased to inform you that your manuscript has been deemed suitable for publication in PLOS ONE. Congratulations! Your manuscript is now being handed over to our production team.

Kind regards, 

on behalf of

Prof. Thomas Behrens 

Academic Editor

PLOS ONE